# Exercising healthy behaviors: A latent class analysis of positive coping during the COVID-19 pandemic and associations with alcohol-related and mental health outcomes

Emma M. McCabe[1], Jeremy W. Luk[2], Bethany L. Stangl[1], Melanie L. Schwandt[2], Ugne Ziausyte[2], Hannah Kim[2], Rhianna R. Vergeer[1], Tommy Gunawan[1,2], Samantha J. Fede[3¤], Reza Momenan[3], Paule V. Joseph[4], David Goldman[2,5], Nancy Diazgranados[2☯], Vijay A. Ramchandani[1☯*]

1 Human Psychopharmacology Laboratory, National Institute on Alcohol Abuse and Alcoholism, Bethesda, MD, United States of America, 2 Office of the Clinical Director, National Institute on Alcohol Abuse and Alcoholism, Bethesda, MD, United States of America, 3 Clinical NeuroImaging Research Core, Office of the Clinical Director, National Institute on Alcohol Abuse and Alcoholism, Bethesda, MD, United States of America, 4 Section of Sensory Science and Metabolism, National Institute on Alcohol Abuse and Alcoholism, Bethesda, MD, United States of America, 5 Laboratory of Neurogenetics, National Institute on Alcohol Abuse and Alcoholism, Rockville, MD, United States of America

☯ These authors contributed equally to this work.
¤ Current address: Department of Psychological Sciences, Auburn University, Auburn, AL, United States of America
* vijayr@mail.nih.gov

## Abstract

### Objective

To identify latent classes of positive coping behaviors during the COVID-19 pandemic and examine associations with alcohol-related and mental health outcomes across participants with and without a history of alcohol use disorder (AUD).

### Methods

Baseline data from 463 participants who were enrolled in the NIAAA COVID-19 Pandemic Impact on Alcohol (C19-PIA) Study were analyzed. Latent class analysis (LCA) was applied to five positive coping behaviors during COVID-19: taking media breaks, taking care of their body, engaging in healthy behaviors, making time to relax, and connecting with others. Latent class differences and the moderating role of history of AUD on six alcohol-related and mental health outcomes were examined using multiple regression models.

### Results

LCA revealed two latent classes: 83.4% High Positive Coping and 16.6% Low Positive Coping. Low Positive Coping was associated with higher levels of perceived stress, anxiety symptoms, and loneliness. A history of AUD was consistently associated with higher levels of alcohol-related and mental health outcomes. Significant interactions between Coping Latent Classes and history of AUD indicated that the associations of Low Positive Coping

**Data Availability Statement:** Data cannot be shared publicly because of Institutional Review Board restrictions. Data are available from the Principal Investigators/Senior Authors (Dr. Vijay Ramchandani [vijayr@nih.gov] or Dr. Nancy Diazgranados [nancy.diazgranados@nih.gov]) or from Sumedha Chawla [sumedha.chawla@nih.gov], after IRB approval, for researchers who meet the criteria for access to deidentified data.

**Funding:** This study was supported by National Institute on Alcohol Abuse and Alcoholism Division of Intramural Clinical and Biological Research (Grants Z1A AA000130 and Z1A AA000466) and a National Institute of Allergy and Infectious Diseases Intramural Targeted Anti-COVID Award to ND and VAR. The funders had no role in study design, data collection and analysis, decision to publish, or preparation of the manuscript.

**Competing interests:** The authors have declared that no competing interests exist.

with problematic alcohol use, depressive symptoms, and drinking to cope motives were either stronger or only significant among individuals with a history of AUD.

## Conclusions

Individuals with a history of AUD may be particularly vulnerable to depressive symptoms and alcohol-related outcomes, especially when they do not utilize positive coping strategies. The promotion of positive coping strategies is a promising avenue to address alcohol-related and mental health problems during a public health crisis and warrants future research.

## Introduction

A combination of healthy lifestyle behaviors, like engaging in regular physical activity, eating a healthy diet, and getting adequate amounts of sleep, has been shown to boost mental health, prevent disease, and prolong life [1–4]. Unhealthy lifestyle behaviors are associated with increased mortality in the U.S. [5–7]. Aside from ability or motivation, stress is a commonly reported reason for lack of engagement with preventative health behaviors and for endorsement of negative lifestyle behaviors, including substance use [7–10].

Alcohol Use Disorder (AUD) is a psychiatric condition affecting over 14.1 million U.S. adults [11]. It is the third leading preventable cause of death in the U.S., costing approximately $249 billion in expenses each year [12, 13]. Perceived stress and anxiety have been linked to greater risk for alcohol problems and relapses for those with a history of problematic alcohol use [10, 14–19]. Individuals with a history of AUD may be more vulnerable during times of high stress as their positive coping skills may be underdeveloped, leading to the adoption of negative coping strategies and unhealthy lifestyle behaviors.

The COVID-19 pandemic is a major public health crisis that has placed considerable stress on individuals within the U.S. and globally [20–22]. The pandemic has resulted in significant lifestyle changes among individuals in the general population, for better or for worse. Changes in time availability, financial stability, and psychological distress have exacerbated poor adherence to healthy behavior guidelines [23, 24]. Much of the emerging research around COVID-19 impact has studied the negative health-related consequences of lockdown and upticks in endorsement of negative coping behaviors [8, 23, 25–28]. Pandemic-related stress has been associated with reports of elevated loneliness, mental health problems, and drinking to cope [20–22, 25, 29–31]. Individuals who adopted or maintained a healthier lifestyle during the pandemic were more likely to experience better physical and mental health outcomes [8, 23, 32].

Although the link between stress and alcohol behavior during the pandemic has been well-documented, relatively few studies have examined the utilization of health-promoting behaviors and their impact on mental health, especially among individuals with a history of AUD [17, 33, 34]. Positive coping behaviors are self-initiated behaviors used to handle psychological distress that, in addition to providing some emotional regulation in the short-term, benefit one's physical and mental well-being over time [35, 36]. In a study conducted in Spain and Brazil between April and May of 2020, De Boni and colleagues found that risky drinking was associated with multiple lifestyle domains, such as diet, physical activity, sleep, social support, and environmental behaviors, highlighting the importance of understanding positive lifestyle factors in mental health and alcohol-related outcomes [37].

Latent class analysis (LCA) is a person-centered approach that has been utilized to examine distinct patterns of COVID-related stressors and experiences, including work-related

interruptions, changes to home life, experiences of social isolation, and financial stress [22, 38, 39]. There are several advantages to this analytic approach [40, 41]. First, LCA is an empirical approach that identifies subgroups of individuals who share similar characteristics based on an underlying latent construct, such as COVID-related stress or COVID-related coping. The application of LCA can help researchers understand the degree to which different types of stressors or coping strategies co-occur among individuals and allow for the opportunity to examine their cumulative effects on external variables. Second, the identification of subgroups of individuals who are at varying levels of risk for mental health and alcohol-related problems can inform targeted outreach and intervention among those most affected. Third, LCA can help integrate information from several observable indicators and capture patterns of endorsement effectively. As a result, more integrated analyses can be conducted using the latent classes rather than the individual indicators. Despite these advantages, LCA has not been applied to examine positive coping behaviors in the context of the COVID-19 pandemic.

To address this literature gap, we conducted a LCA of positive coping behaviors during the COVID-19 pandemic to identify subgroups of individuals who may exhibit multiple positive coping strategies. First, latent class differences in endorsement of individual negative coping behaviors were evaluated. Second, latent class differences in alcohol-related behaviors, perceived stress, mental health, and loneliness were assessed using validated measures. Third, history of AUD was tested as a potential moderator between latent class membership and clinical outcomes. We hypothesized that any latent class characterized by lower positive coping would be associated with negative coping behaviors, as well as higher scores on alcohol-related and mental health outcomes. We further hypothesized that the utilization of positive coping strategies may be particularly useful among those with a history of AUD.

## Methods

### Participants

Between June 3, 2020 and January 6, 2022, 463 participants completed the baseline survey of the National Institute on Alcohol Abuse and Alcoholism (NIAAA) COVID-19 Pandemic Impact on Alcohol (C19-PIA) Study. Details about the study design and participant recruitment procedures can be found in prior C19-PIA Study publications [22, 42]. In brief, individuals who previously participated in research at NIAAA via the Natural History Protocol (NHP) were invited to join a study focused on the impact of the pandemic on alcohol-related behaviors [43]. Participants provided verbal consent at the start of the initial phone survey and consent was documented as part of the survey. Participants were provided a copy of the consent document along with their initial online survey. The baseline survey consisted of two parts, with the first part conducted over the phone, and the second part conducted either over the phone or online. Participants who completed the first part were compensated with $25, and those who completed both parts were compensated with $50. Based on data from Structured Clinical Interview for DSM-IV or DSM-5 collected as part of the NHP, 41.9% ($n$ = 194) of participants met criteria for an AUD diagnosis. At the time of the C19-PIA Study, 84.7% of participants resided in either Maryland, Virginia or Washington, D.C. Given rolling enrollment in the C19-PIA Study, participants were enrolled at different stages of the pandemic. Four pandemic phases were created based on frequency distributions of enrollment timing while considering COVID-19 infection level and state policies in the Washington D.C. Metro area: Phase 1 (June 3 to July 31, 2020); Phase 2 (August 1 to November 22, 2020); Phase 3 (November 23, 2020 to February 28, 2021); and Phase 4 (March 1, 2021 to January 6, 2022). The C19-PIA Study protocol was approved by the NIH Intramural Institutional Review Board and

is registered in clinicaltrials.gov (NCT04391816). Authors that were involved in recruitment and data collection had access to identifiable information about participants.

## Measures

**Alcohol Use Disorders Identification Test (AUDIT)** is a 10-item questionnaire that assesses for alcohol consumption and problematic use and is a validated screening tool for AUD [44]. In this study, the total AUDIT score was used to capture level of problematic alcohol use (Cronbach's α = 0.94).

**Perceived Stress Scale (PSS)** is a widely used 10-item scale to measure the perception of stress [45]. In this study, the Cronbach's α for PSS was 0.92.

**Generalized Anxiety Disorder Scale-7 (GAD-7) and Patient Health Questionnaire-9 (PHQ-9)** are validated self-administered questionnaires to measure the severity of anxiety and depression symptoms [46, 47]. In this study, the Cronbach's αs for GAD-7 and PHQ-9 were 0.92 and 0.89, respectively.

**UCLA Loneliness Scale (UCLA-LS)** is a 20-item scale designed to measure a person's subjective feeling of loneliness and social isolation in the last month [48]. In this study, the Cronbach's α for UCLA-LS was 0.97.

**Drinking Motives Questionnaire (DMQ-R)** measures motives for alcohol consumption and consists of four subscales: social motives, coping motives, enhancement motives, and conformity motives [49]. In this study, the DMQ-R was only administered to participants who endorsed any drinking behavior during the pandemic. Moreover, as the focus of this investigation was on coping behaviors in the context of the COVID-19 pandemic, only the coping motives subscale (Cronbach's α = 0.92) was analyzed.

**The COVID-19 Community Survey Question Bank** developed by the Centers for Disease Control and Prevention contains items related to coping behaviors during the COVID-19 pandemic [50]. These items were adapted in the current study and participants were asked: "As a result of the COVID-19 pandemic, are you doing any of the following?" Participants were provided with a list of six positive coping behaviors and 11 negative coping behaviors. For positive coping behaviors, the item "Contacting a healthcare provider" was excluded from the analyses because the need and appropriateness of this behavior depends on individual circumstances and may not be broadly applicable. The remaining five positive coping behaviors chosen to represent "positive coping" had moderate-to-high levels of endorsement and could be easily interpreted as positive in most contexts. As for negative coping behaviors, we excluded items that were infrequently endorsed (e.g., use of prescription and non-prescription drugs, < 5%) and focused on seven items related to sleep, smoking cigarettes, drinking alcohol, and eating behaviors. For all items, participants were asked to report endorsement using a dichotomous "yes" or "no" response scheme. The wording of the five positive coping behaviors that were treated as indicators in the latent class analysis can be found in the figure caption to Fig 1.

## Statistical analysis

Latent class analysis (LCA) was conducted to identify subgroups with distinct patterns of positive coping behavior endorsement. Model fit statistics and interpretability of latent classes were considered in selecting the optimal number of classes [40]. Chi-square tests were conducted to evaluate the associations between latent class membership and individual negative coping behaviors. Multiple regression models adjusting for age, gender, race, ethnicity, marital status, income level, and pandemic phase were used to examine latent class differences in problematic alcohol use (AUDIT), perceived stress (PSS), anxiety symptoms (GAD-7), depressive symptoms (PHQ-9), loneliness (UCLA-LS), and drinking to cope (DMQ Coping). The

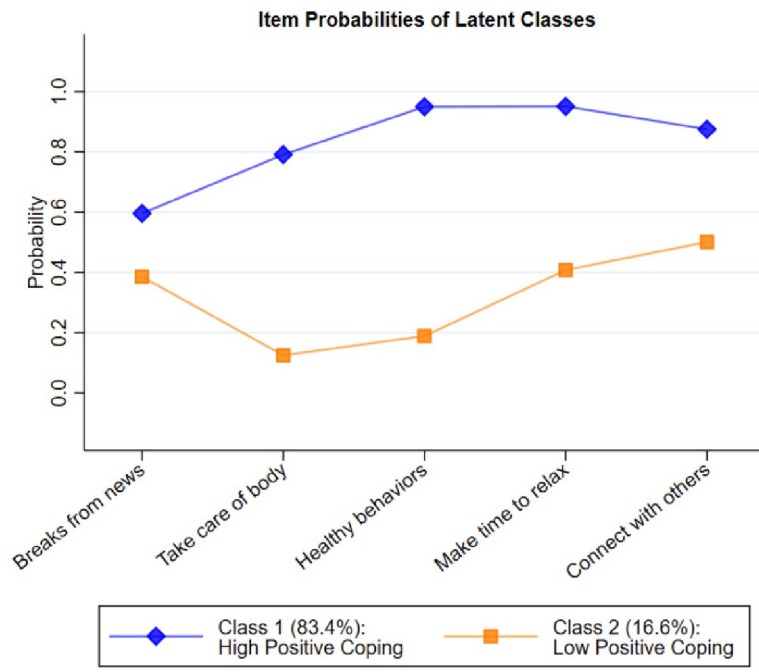

**Item Probabilities of Latent Classes**

Note: Class 1 had moderate-to-high probabilities of endorsing positive coping behaviors, while Class 2 had low probabilities of endorsement. The positive coping behaviors are as follows:
"Taking breaks from watching, reading, or listening to news stories, including social media."
"Taking care of your body, such as taking deep breaths, stretching, or meditating."
"Engaging in healthy behaviors like trying to eat healthy, exercising regularly, getting plenty of sleep or avoiding alcohol or drugs."
"Making time to relax."
"Connecting with others, including talking with people you trust about your concerns and how you are feeling."

**Fig 1. Item probabilities of latent class endorsement for the five positive coping behaviors.**

interaction term was created using centered Positive Coping Latent Classes (0 = High Positive Coping and 1 = Low Positive Coping) and centered History of AUD (0 = no history of AUD; 1 = history of AUD). For significant interaction effects, simple slope analyses were conducted to decompose the conditioned effect of Positive Coping on outcomes by History of AUD. For nonsignificant interaction effects, the final models were re-estimated after removing the interaction term [51].

For the first four outcomes, they were assessed by trained clinicians over the phone and there were no missing data. For the last two outcomes, loneliness and drinking to cope, there were missing data as they were assessed in an optional web-based survey as the second part of the baseline assessment [22]. The number of valid observations for loneliness and drinking to cope were 366 (79%) and 310 (67%), respectively. Missing data for drinking to cope was partially due to a skip logic for nondrinkers ($n$ = 48). All other missing data were handled using multiple imputation with chained equations under the assumption of missing at random [52].

The number of imputations was 20 and specification of chained equations was performed based on characteristics of the imputed variable (e.g., ordered logistic regression for income, and linear regressions for continuous variables). Preliminary analyses were conducted in SPSS 28 (IBM Corp., Armonk, NY). The LCA was performed using MPlus 8.4 (Muthen & Muthen, 1998–2019) and the multiple regression models with multiple imputation were conducted using Stata 17 (StataCorp LLC, College Station, TX).

## Results

### Sample characteristics

Table 1 shows sample characteristics for the overall sample and by latent class. The study sample had a mean age of 44.8 (SD = 14.2) and was diverse in terms of sex, race, and ethnicity. History of AUD was positive among 194 participants (41.9%). Most sample characteristics did not significantly differ between latent classes. Chi-square tests run on marital status and pandemic phase by latent class were statistically significant, suggesting that positive coping varied by

**Table 1. Sample characteristics for the overall sample and by latent class.**

| Variable | Overall Sample (*n* = 463, 100%) | | High-Positive Coping (*n* = 386, 83.4%) | | Low-Positive Coping (*n* = 77, 16.6%) | | Chi-Square Tests | |
|---|---|---|---|---|---|---|---|---|
| | Frequency | Percent | Frequency | Percent | Frequency | Percent | Chi-sq | *p* |
| **Age (years)** | | | | | | | | |
| 18–34 | 158 | 34.1 | 130 | 33.7 | 28 | 36.4 | 0.206 | 0.902 |
| 35–54 | 162 | 35.0 | 136 | 35.2 | 26 | 33.8 | | |
| ≥55 | 143 | 30.9 | 120 | 31.1 | 23 | 29.9 | | |
| **Sex** | | | | | | | | |
| Female | 238 | 51.4 | 197 | 51.0 | 41 | 53.3 | 0.126 | 0.723 |
| Male | 225 | 48.6 | 189 | 49.0 | 36 | 46.8 | | |
| **Race** | | | | | | | | |
| White | 234 | 50.5 | 197 | 51.0 | 37 | 48.1 | 0.277 | 0.871 |
| Black | 157 | 33.9 | 129 | 33.4 | 28 | 36.4 | | |
| Other | 72 | 15.6 | 60 | 15.5 | 12 | 15.6 | | |
| **Ethnicity** | | | | | | | | |
| Not Hispanic or Latino | 412 | 89.0 | 341 | 88.3 | 71 | 92.2 | 1.017 | 0.602 |
| Hispanic or Latino | 36 | 7.8 | 32 | 8.3 | 4 | 5.2 | | |
| **AUD History** | | | | | | | | |
| Non-AUD | 269 | 58.1 | 228 | 59.1 | 41 | 53.3 | 0.893 | 0.345 |
| AUD | 194 | 41.9 | 158 | 40.9 | 36 | 46.8 | | |
| **Marital Status** | | | | | | | | |
| Single | 294 | 65.0 | 244 | 65.1 | 50 | 64.9 | 9.271 | **0.010** |
| Married | 94 | 20.8 | 85 | 22.7 | 9 | 11.7 | | |
| Other | 64 | 14.2 | 46 | 12.3 | 18 | 23.4 | | |
| **Income Level** | | | | | | | | |
| <$20,000 | 114 | 25.3 | 90 | 23.9 | 24 | 32.4 | 4.679 | 0.096 |
| $20,000-$74,999 | 195 | 43.2 | 161 | 42.7 | 34 | 46.0 | | |
| ≥$75,000 | 142 | 31.5 | 126 | 33.4 | 16 | 21.6 | | |
| **Pandemic Phase** | | | | | | | | |
| Phase 1 | 105 | 22.7 | 76 | 19.7 | 29 | 37.7 | 13.757 | **0.003** |
| Phase 2 | 140 | 30.2 | 126 | 32.6 | 14 | 18.2 | | |
| Phase 3 | 127 | 27.4 | 107 | 27.7 | 20 | 26.0 | | |
| Phase 4 | 91 | 19.7 | 77 | 20.0 | 14 | 18.2 | | |

*Note*: The following were included in "other" marital status: Divorced (*n* = 43), Separated (*n* = 11), Widowed (*n* = 8), Other Marital Status (*n* = 2). Participants were enrolled in the C19-PIA study in four pandemic phases: Phase 1 (June 3 to July 31, 2020); Phase 2 (August 1 to November 22, 2020); Phase 3 (November 23, 2020 to February 28, 2021); and Phase 4 (March 1, 2021 to January 6, 2022). Missing data was recorded for ethnicity (*n* = 15, 3.2%), marital status (*n* = 11, 2.4%) and income level (*n* = 12, 2.6%). Significant differences are highlighted in bold.

marital status and pandemic phase. Specifically, compared to participants with High Positive Coping, participants with Low Positive Coping were less likely to be "married" (11.7% vs 22.7%) and more likely to report "other" marital status (e.g., divorced, separated, widowed; 23.4% vs 12.3%). Low Positive Coping was more frequently endorsed during the early phase of the pandemic (Phase 1, 37.7%), whereas High Positive Coping was more frequently endorsed from August 1 to November 22, 2020 (Phase 2, 32.6%). Though not statistically significant, the percentages of participants with household income at $75,000 or higher were 33.4% in the High Positive Coping Class and 21.6% in the Low Positive Coping Class.

## Latent class analysis

The two-class solution yielded optimal model fit in the LCA (Table 2). A two-class model was determined to be optimal since it had the lowest BIC score (2353.999) and both likelihood tests were significant ($p < 0.001$) when comparing two classes to one class only. Participants were classified into two classes based on their patterns of positive coping behavior endorsement. Class 1 had moderate-to-high probabilities of endorsing positive coping behaviors (High Positive Coping, 83.4%). Class 2 had low probabilities of endorsement (Low Positive Coping, 16.6%).

Fig 1 depicts probabilities of endorsement for each positive coping behavior by latent class. Participants in the High Positive Coping Class had the highest probability of endorsing "making time to relax" (95.1%) and had the lowest probability of endorsing "taking media breaks" (59.6%), whereas those in the Low Positive Coping Class had the highest probability of endorsing "connecting with others" (50.1%) and had the lowest probability of endorsing "taking care of your body" (12.5%). Endorsement probability differed by at least 20 percent between High Positive Coping and Low Positive Coping Classes across all positive coping behaviors, with the largest gap in endorsement probability occurring with "engaging in healthy behaviors" (95.0% in the High Positive Coping Class and 18.9% in the Low Positive Coping Class).

## Latent class associations with negative coping behaviors

After conducting the LCA, we first examined if the percentages of negative coping behaviors during the pandemic differed by latent class membership. Table 3 shows the percentages of positive endorsement for seven negative coping behaviors: getting more sleep than usual, getting less sleep than usual, smoking cigarettes more or vaping more, drinking alcohol, eating high fat or sugary foods, eating more than usual, and eating less than usual.

A higher proportion of participants in the High Positive Coping class relative to participants in the Low Positive Coping class reported getting more sleep than usual ($p = 0.001$). Conversely, a higher proportion of participants in the Low Positive Coping class relative to participants in the High Positive Coping class reported getting less sleep than usual

**Table 2. Model fit indices for the latent class models.**

| Criteria | Number of Latent Classes | | | | |
| --- | --- | --- | --- | --- | --- |
| | 1 | 2 | 3 | 4 | 5 |
| Loglikelihood | -1243.309 | -1143.242 | -1136.943 | -1134.760 | -1132.086 |
| Akaike Information Criterion (AIC) | 2496.617 | 2308.484 | 2307.886 | 2315.521 | 2322.172 |
| Bayesian Information Criterion (BIC) | 2517.306 | 2353.999 | 2378.227 | 2410.668 | 2442.166 |
| Sample-Size Adjusted BIC | 2501.437 | 2319.087 | 2324.274 | 2337.692 | 2350.127 |
| Entropy | N/A | 0.840 | 0.664 | 0.693 | 0.925 |
| Lo-Mendell-Rubin Likelihood Ratio Test | N/A | 0.0000 | 0.2733 | 0.4551 | 0.4388 |
| Bootstrapped Likelihood Ratio Test | N/A | 0.0000 | 0.0769 | 1.0000 | 1.0000 |

**Table 3. Endorsement of negative coping behaviors by positive coping latent classes.**

| Variable | High Positive Coping | Low Positive Coping | Chi Square Statistic | p |
|---|---|---|---|---|
| Getting more sleep than usual | **33.9%** | 15.6% | 10.13 | **0.001** |
| Getting less sleep than usual | 24.4% | **44.2%** | 12.59 | **<0.001** |
| Smoking more cigarettes or vaping more | 12.2% | **24.7%** | 8.21 | **0.004** |
| Drinking alcohol | 29.3% | **54.6%** | 18.41 | **<0.001** |
| Eating high fat or sugary foods | 40.4% | 52.0% | 3.50 | 0.06 |
| Eating more food than usual | 24.9% | **36.4%** | 4.32 | **0.04** |
| Eating less food than usual | 15.5% | 23.4% | 2.81 | 0.09 |

*Note*: "Getting more sleep than usual" was included as a negative coping behavior because excessive sleep can be a symptom of mental health disorders like depression [53, 54]. Significant differences are highlighted in bold.

($p < 0.001$). Similarly, individuals with Low Positive Coping were more likely than individuals with High Positive Coping to report other negative coping behaviors, including smoking cigarettes or vaping more ($p = 0.004$), drinking alcohol ($p < 0.001$), and eating more food than usual ($p = 0.04$).

## Latent class differences in alcohol-related and mental health outcomes and moderation by history of AUD

Fig 2 shows the pattern of results regarding the associations between Positive Coping Latent Classes and outcomes of interest by history of AUD. There were significant interaction effects between Low Positive Coping and history of AUD on alcohol problems (Panel A), depressive symptoms (Panel D), and drinking to cope (Panel F). The overall main effects of Low Positive Coping were statistically significant for perceived stress (Panel B), anxiety symptoms (Panel C), and loneliness (Panel E). The main effects of AUD were statistically significant for all clinical outcomes (Panels A-F).

Full results from multiple regression analyses modeling the main and interaction effects of Low Positive Coping Class and history of AUD on these six outcomes after adjusting for covariates can be found in S1 Table. Table 4 summarizes the key interaction and main effects. A history of AUD moderated the associations between Low Positive Coping and three outcomes. First, Low Positive Coping was associated with higher problematic alcohol use among individuals with a history of AUD ($b = 8.35$, 95% CI = 5.31, 11.39), but not among individuals without a history of AUD ($b = 2.15$, 95% CI = -0.64, 4.95). Second, Low Positive Coping was more strongly associated with higher depressive symptoms among individuals with a history of AUD ($b = 6.24$, 95% CI = 4.35, 8.13) than among individuals without a history of AUD ($b = 2.80$, 95% CI = 1.07, 4.54). Third, Low Positive Coping was associated with drinking to cope among individuals with a history of AUD ($b = 4.09$, 95% CI = 2.17, 6.00), but not among individuals without a history of AUD ($b = 0.47$, 95% CI = -1.35, 2.30). In terms of overall main effects, Low Positive Coping was positively associated with higher perceived stress ($b = 4.52$, 95% CI = 2.47, 6.56), higher anxiety symptoms ($b = 3.47$, 95% CI = 2.27, 4.67), and higher loneliness ($b = 5.30$, 95% CI = 0.78, 9.82). Moreover, a history of AUD was consistently associated with higher levels of alcohol-related and mental health outcomes.

## Discussion

In this study, we assessed positive coping behaviors during the COVID-19 pandemic and examined the association between patterns of coping behavior endorsement and alcohol-

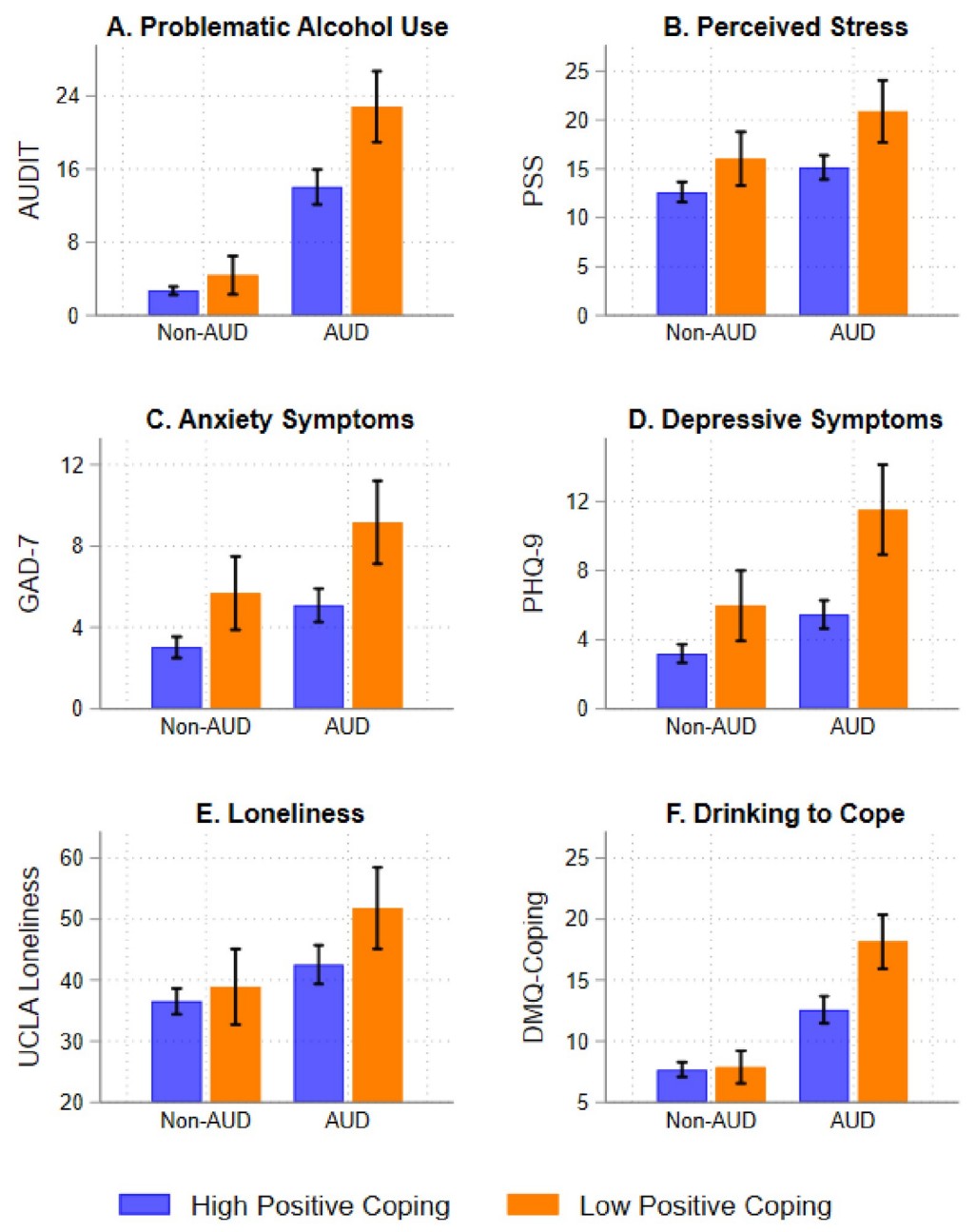

**Fig 2. Alcohol-related and mental health outcomes by latent classes and history of AUD.**

**Table 4. Multiple regression analyses modeling the main and interaction effects of coping classes and history of AUD on alcohol-related and mental health outcomes.**

| | AUDIT (n = 463) | | | PSS (n = 463) | | | GAD-7 (n = 463) | | |
|---|---|---|---|---|---|---|---|---|---|
| Variable | b | SE | p | b | SE | p | b | SE | p |
| Low Positive Coping | 2.154 | 1.421 | 0.130 | 4.518 | 1.040 | <**0.001** | 3.474 | 0.611 | <**0.001** |
| AUD | 11.348 | 0.912 | <**0.001** | 2.605 | 0.824 | **0.002** | 2.060 | 0.485 | <**0.001** |
| Low Positive Coping * AUD | 6.200 | 2.082 | **0.003** | – | – | – | – | – | – |
| | PHQ-9 (n = 463) | | | UCLA-LS (n = 366) | | | DMQ-Coping (n = 310) | | |
| Variable | b | SE | p | b | SE | p | b | SE | p |
| Low Positive Coping | 2.803 | 0.883 | **0.002** | 5.302 | 2.294 | **0.022** | 0.472 | 0.926 | 0.611 |
| AUD | 2.068 | 0.567 | <**0.001** | 6.587 | 1.884 | **0.001** | 4.777 | 0.668 | <**0.001** |
| Low Positive Coping * AUD | 3.435 | 1.294 | **0.008** | – | – | – | 3.616 | 1.295 | **0.006** |

*Note*: Full multiple regression results with all covariate effects can be found in S1 Table. All multiple regression analyses adjusted for age, sex, race, ethnicity, marital status, income level, and pandemic phase. Because the Coping * AUD interaction was not significant for perceived stress, anxiety symptoms, and loneliness (see S1 Table), the interaction term was dropped in the final models presented here. The sample sizes listed in this Table indicate the number of responses for each outcome variable. There were no missing data in problematic alcohol use, perceived stress, anxiety symptoms, and depressive symptoms (n = 463). There were missing data in loneliness (missing n = 97), drinking to cope (missing n due to skip logic for non-drinkers = 48; other missing n = 105), marital status (missing n = 11), and income level (missing n = 12). Missing data due to skip logic were listwise deleted. Missing data not due to skip logic were handled using multiple imputation. All significant findings reported in this Table were replicated in multiple regression models without multiple imputation. Significant differences are highlighted in bold.

related and mental health outcomes. Most participants (83.4% High Positive Coping) had moderate-to-high probabilities of endorsing multiple positive coping behaviors. Participants in the Low Positive Coping Class (16.6%) were more likely than participants in the High Positive Coping Class to endorse negative coping behaviors like getting less sleep than usual, smoking more cigarettes or vaping more, drinking alcohol, and eating more food than usual. Participants with a history of AUD had significantly greater scores on all clinical outcomes compared to those with no history of AUD, consistent with prior research on the adverse impacts of AUD on physical and mental health [13, 20, 54, 55]. History of AUD further moderated the associations of Positive Coping Class with alcohol problems, depressive symptoms, and drinking to cope motives. Specifically, participants in the Low Positive Coping Class with a history of AUD had the greatest scores on these measures compared to all other groups, highlighting the importance of positive coping as a protective factor among individuals with AUD.

Individuals may turn to positive or negative health behaviors to cope during the pandemic [7, 21, 24]. The majority of our sample were classified into the High Coping Latent Class that was characterized by endorsement of multiple positive coping behaviors. Often, engagement with one positive coping behavior is accompanied by or reinforces engagement with other positive coping behaviors [56, 57]. For example, older adults who maintained social connection during the pandemic were able to maintain regular routines and achieve better mental health outcomes [58]. As several pandemic studies have explored, increased boredom, psychological distress, and time availability contribute to negative coping choices [24, 29]. To counter these negative choices, the promotion of positive coping strategies can help individuals occupy time with meaningful activities, with less space for negative thinking and unhealthy coping behaviors [24, 58, 59].

Healthy lifestyle behaviors have been proposed as effective strategies to increase self-regulation and coping capacities in face of stress exposure [7]. During the COVID-19 pandemic,

psychological distress attributable to social isolation was described as a possible reason why individuals experienced altered eating habits, disrupted sleep, and reduced physical activities [8]. Consistent with these theoretical contributions, our study showed that individuals who utilized more positive coping behaviors were less likely to engage in negative lifestyle coping behaviors. Moreover, individuals with a history of AUD were particularly vulnerable to increased problematic drinking, depressive symptoms, and drinking to cope if they also did not utilize positive coping. These significant interaction effects highlight the need to support individuals with AUD to better manage symptoms of depression and develop alternative coping strategies, so that they do not turn to drinking for tension reduction or short-term psychological relief from negative emotions [19, 60–62].

In line with previous literature, this study suggests positive coping behaviors may be generally protective against perceived stress, anxiety symptoms, and loneliness [63–65]. These associations did not vary significantly by history of AUD, suggesting that positive coping can be beneficial for stress and anxiety management as well as prevention of social isolation regardless of history of AUD. Of note, the positive coping behaviors assessed in this study included a range of coping options that can be incorporated into time-limited, brief alcohol-related and mental health interventions. To further inform practice, psychological factors that contribute to the likelihood of endorsing positive health behaviors, like self-efficacy or motivation, should be considered along with existing theories of health behavior change (i.e., Self-Determination Theory) [66, 67].

To best inform targeted dissemination of positive coping strategies, it is important to understand whether segments of our society may be less likely to utilize positive coping. In our study, we found that low positive coping was more frequently endorsed by individuals who were divorced, separated, or widowed. This group of individuals may lack support from their family to engage in positive coping, or they may have less free time or lower financial resources to support certain types of positive coping (e.g., exercising regularly or making time to relax) [68–70]. Community-based support groups for these individuals may help encourage positive coping behaviors [71]. Prior literature also suggests that individuals with higher socioeconomic status may have greater access to resources like money, time, and social support to promote healthy lifestyle behaviors [29, 72, 73]. Although we did not find significant differences in income level by latent classes, in theory, individuals with greater financial resources would have access to more positive coping activities (e.g., joining a gym membership or buying healthier food). More research is needed to examine how financial and social capital can used to help promote positive coping behaviors.

This study has several limitations. First, the cross-sectional design prohibited causal interpretation. For example, while positive coping strategies can be used to manage symptoms of anxiety and depression, anxiety and depression can also lead to lower use of positive coping as individuals with anxiety or depression may be more prone to avoidant behavior or social isolation [74–76]. Longitudinal research is needed to disentangle the directionality of effects. Second, the "engaging in healthy behaviors" question was broad and listed several health behaviors (Fig 1), so participants' interpretation of this question could differ and introduce measurement bias. Third, for clinical variables assessed by phone, social desirability could lead to underreporting of symptoms. Fourth, retrospective self-report of coping behaviors and clinical outcomes may be subject to recall bias. Fifth, although pandemic phases were controlled for as a covariate in the analyses, shifts in pandemic-related mandates related to quarantining and social distancing may have influenced levels of stress, time availability, and motivation to engage in positive or negative coping behaviors, as well as the assessed outcomes [24, 64, 77, 78]. Finally, missing data in loneliness and drinking to cope may have biased the results of these analyses.

Despite these limitations, this study is among the first to utilize a person-centered approach to capture multiple positive coping behaviors during the COVID-19 pandemic. Analyses of self-reported coping behaviors, drinking motives and behaviors, perceived stress, mental health, and loneliness support that positive coping may mitigate alcohol problems and mental health symptoms, especially for those with a history of AUD. Additional research on positive coping behaviors may inform the development of strength-based approaches to address the mental health and alcohol-related burden during the pandemic and beyond.

## Supporting information

**S1 Table. Full results from multiple regression models with covariates and coping by AUD interaction terms.**
(DOCX)

## Acknowledgments

The authors would like to thank Sumedha Chawla, Beth Lee, Megan Carraco, Sheila Walsh, Betsy Davis, Cheryl Jones, Alyssa Brooks, Tonette Vinson, Yvonne Horneffer, LaToya Sewell, the ClinDB IT team (Thuy Van, Etienne Lamoreaux, Denise Gates-Nee, Nancy Agarwal, Patty Bates, Jonathan Folkers), and the intrepid postbaccalaureate Intramural Research Training Award fellows (Jared Axelowitz, Noa Leiter, Carlos Melendez, James Morris, Kurren Parida) for supporting the COVID-19 Pandemic Impact on Alcohol Study (C19-PIA Study).

## Author Contributions

**Conceptualization:** Emma M. McCabe, Samantha J. Fede, Reza Momenan, Paule V. Joseph, David Goldman, Nancy Diazgranados, Vijay A. Ramchandani.

**Data curation:** Melanie L. Schwandt.

**Formal analysis:** Emma M. McCabe, Jeremy W. Luk, Bethany L. Stangl, Hannah Kim, Rhianna R. Vergeer.

**Funding acquisition:** Nancy Diazgranados, Vijay A. Ramchandani.

**Investigation:** Emma M. McCabe, Jeremy W. Luk, Ugne Ziausyte, Vijay A. Ramchandani.

**Methodology:** Jeremy W. Luk, Bethany L. Stangl, Hannah Kim, Rhianna R. Vergeer, Tommy Gunawan, Samantha J. Fede, Paule V. Joseph, Vijay A. Ramchandani.

**Project administration:** Nancy Diazgranados, Vijay A. Ramchandani.

**Resources:** David Goldman, Nancy Diazgranados, Vijay A. Ramchandani.

**Supervision:** Bethany L. Stangl, Vijay A. Ramchandani.

**Validation:** Melanie L. Schwandt.

**Visualization:** Jeremy W. Luk.

**Writing – original draft:** Emma M. McCabe, Jeremy W. Luk.

**Writing – review & editing:** Jeremy W. Luk, Bethany L. Stangl, Melanie L. Schwandt, Ugne Ziausyte, Hannah Kim, Rhianna R. Vergeer, Tommy Gunawan, Samantha J. Fede, Reza Momenan, Paule V. Joseph, David Goldman, Nancy Diazgranados, Vijay A. Ramchandani.

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
