## [Decision Letter · Decision Letter 0]

7 Jun 2023

PONE-D-23-11488Exercising healthy behaviors: A latent class analysis of positive coping during the COVID-19 pandemic and associations with alcohol-related and mental health outcomesPLOS ONE

Dear Dr. Ramchandani,

Thank you for submitting your manuscript to PLOS ONE. After careful consideration, we feel that it has merit but does not fully meet PLOS ONE’s publication criteria as it currently stands. Therefore, we invite you to submit a revised version of the manuscript that addresses the points raised during the review process.

We look forward to receiving your revised manuscript.

Kind regards,

Giuseppe Marano

Academic Editor

PLOS ONE

“The authors would like to thank Sumedha Chawla, Beth Lee, Megan Carraco, Sheila Walsh, Betsy Davis, Cheryl Jones, Alyssa Brooks, Tonette Vinson, Yvonne Horneffer, LaToya Sewell, the ClinDB IT team (Thuy Van, Etienne Lamoreaux, Denise Gates-Nee, Nancy Agarwal, Patty Bates, Jonathan Folkers), and the intrepid postbaccalaureate Intramural Research Training Award fellows (Jared Axelowitz, Noa Leiter, Carlos Melendez, James Morris, Kurren Parida) for supporting the COVID-19 Pandemic Impact on Alcohol Study (C19-PIA Study). This study was supported by National Institute on Alcohol Abuse and Alcoholism Division of Intramural Clinical and Biological Research (Grants Z1A AA000130 and Z1A AA000466) and a National Institute of Allergy and Infectious Diseases Intramural Targeted Anti-COVID Award to Nancy Diazgranados and Vijay A. Ramchandani.”

“This study was supported by National Institute on Alcohol Abuse and Alcoholism Division of Intramural Clinical and Biological Research (Grants Z1A AA000130 and Z1A AA000466) and a National Institute of Allergy and Infectious Diseases Intramural Targeted Anti-COVID Award to ND and VAR. The funders had no role in study design, data collection and analysis, decision to publish, or preparation of the manuscript.”

Reviewers' comments:

Reviewer's Responses to Questions

**Comments to the Author**

1. Is the manuscript technically sound, and do the data support the conclusions?

Reviewer #1: Yes

Reviewer #2: Yes

Reviewer #3: Partly

2. Has the statistical analysis been performed appropriately and rigorously? 

Reviewer #1: Yes

Reviewer #2: Yes

Reviewer #3: I Don't Know

3. Have the authors made all data underlying the findings in their manuscript fully available?

Reviewer #1: No

Reviewer #2: Yes

Reviewer #3: Yes

4. Is the manuscript presented in an intelligible fashion and written in standard English?

Reviewer #1: Yes

Reviewer #2: Yes

Reviewer #3: Yes

5. Review Comments to the Author

Reviewer #1: Summary: This paper utilizes a latent class analysis approach to identify distinct groups of individuals based on their positive coping behaviors and investigates the association between these groups and both alcohol-related and mental health outcomes, while considering the presence or absence of a history of alcohol use disorder (AUD). Employing a variety of coping behavior responses, the results reveal the existence of two latent classes - namely high positive coping and low positive coping. High positive coping behavior is found to be inversely related to negative coping behaviors. The results indicate that individuals with a history of AUD are more likely to experience alcohol-related and mental health problems. Furthermore, low positive coping behaviors, as well as their interaction with a history of AUD, have significant associations with some of these outcomes.

Overall, I found the paper to be informative and interesting but there are a few areas where revision and further clarification could improve the paper.

Comments:

1. The paper includes a large number of analyses, but for some of them it was not clearly communicated in the paper why certain analyses were conducted. For instance, the analysis that looks at the relationship between positive and negative coping behavior in page 13 (Table 3). It is not clear why this analysis was done. Was it a sanity check of the latent classes or did it have additional purpose? For the sake of clarity and completeness, the authors should explain more clearly the purpose and relevance of each analysis to the research question. This would help readers to follow the logic and importance of each analytical step taken.

2. In the statistical analysis, page 9, the authors talk about pandemic phases, but we do not know what time period constitutes these phases unless we read the caption in Table 1. The authors should discuss the pandemic phases in the main text and what data was collected in which phase.

3. The section “Latent Class Differences in Alcohol-Related and Mental Health Outcomes and Moderation by History of AUD” on page 14 requires more detailed discussion of the results by the different outcome variables in Figure 2.

4. Figure 2 requires a caption explaining what the bar represents (mean/ average of the outcomes) and what the error bars represents (e.g., 95% confidence interval)

5. Table 4 is too crowded. Since the other covariates such as age and gender are not discussed in the paper, the authors can exclude those variables from the table and just show the estimates for low positive coping, AUD and their interaction. The full table with all the covariates can be included in the appendix. The authors should also include the sample size for all the outcome variables.

6. In line 313 on page 19, the sentence talks about the findings of this study but provides citations of other papers. I am not sure why that was done.

7. In the discussion section, from page 19 onwards, the authors make more general statements about how positive coping behavior may be protective against alcohol related and mental health outcomes while the authors findings in the paper were more nuanced. For example, the regression analysis shows that low positive coping is only associated with alcohol related outcomes for individuals with AUD. While the authors’ discussion also focuses on the outcomes that they found a significant relationship with, I think the readers will also benefit from discussion of the outcomes that the authors did not see a significant association with. For instance, in line 322 page 19, the authors cite and discusses a study that found how positive coping behaviors could be used to prevent feelings of distress and loneliness, however, in this study we do not see positive coping behavior to be associated with loneliness.

8. Minor comment: On page 5, SCID-IV and SCID-5 use two different numerals in the name. Unless this is how the names should be written, perhaps either use English or Roman numerals.

Reviewer #2: In this cross-sectional study, authors identify latent classes of coping behaviors during the first year of the COVID-19 pandemic and examine associations with alcohol-related and mental health outcomes among participants with and without alcohol use disorder (AUD). The study used data collected from a subset of participants in an existing NIAAA cohort study and data was collected online and by phone over the first year of the pandemic. Latent class analysis, chi-squares, and regression models were used to analyze results. Results showed that participants with low positive coping had significantly greater alcohol use and poorer mental health outcomes and were more likely to endorse negative coping behaviors.

The manuscript is written clearly and is well-supported by literature. There are a few additional topics to consider for inclusion in the discussion:

It’s unclear to me why the author focuses on “more likely to report being single or married as opposed to ‘other’”. Single versus married seems to be a much more disparate distinction than “other” and there’s no literature to support why marital status may be playing a role in latent classes or behaviors.

There is no discussion of how pandemic phase may play a role in results, although it is highlighted as a participant characteristic in Table 1. We know that recommended behavioral guidelines shifted quite a bit throughout these phases, as did the perceived severity of outbreak as different variants were introduced. Seems like phase would be an important thing to consider when assessing when and why people were employing specific coping strategies and the paper would be significantly strengthened by adding some sort of discussion and literature around this consideration.

While the mention of future directions investigating predictive factors that prime individuals to select coping behaviors is interesting and well cited, suggest that the authors tie this back to their own work and how the demographics of their own sample may or may not reflect the potential predictors they reference in this section of the discussion. How does the makeup of your sample support or not support these findings?

Minor Suggestions:

Relatively minor changes to the organization of the introduction would strengthen its argument. Though all the information is there and well cited, it feels a bit disjointed. Suggest reorganizing the section to discuss general research on the effect of AUD, health behaviors, and stress on the population and then shift to how the pandemic may have exacerbated existing associations rather than switching back and forth between general and pandemic related findings to more clearly delineate arguments

Methods - The author states “Participants included healthy volunteers, moderate social drinkers, and individuals with alcohol use disorder (AUD).” It’s unclear what these categories mean at this point and would suggest deleting this sentence or moving it to later in the methods section when it is more clear what they are referring to.

Results - Please correct formatting in frequency box of the Pandemic Phase section of Table 1. Interesting that there is a clear difference in high and low-coping behaviors during Phases 1 and 2 of the pandemic, but I don’t see that mentioned in the initial results.

Reviewer #3: This manuscript reports an analysis of data from a cross-sectional survey of 463 adults, about half of whom had alcohol use disorder (AUD). Surveys were completed between June 2020 and January 2022, during the COVID-19 pandemic. The authors fit a latent class model to self-reported data on positive coping behaviors, then related class membership to negative coping behaviors, problematic alcohol use, perceived stress, anxiety, depression, loneliness, and drinking to cope.

The authors chose a two-class latent class solution and found that relative to the “high positive coping” class, the “low positive coping” class exhibited more negative coping behaviors, perceived stress, anxiety, and depression. Moderation analyses showed stronger associations between membership in the high vs. low positive coping classes and outcomes for those with history of AUD.

I believe more information is needed about the design and analysis before the manuscript could be described as technically sound.

BIGGEST ISSUES

1. Why was a latent class model used? The few sentences lines 84-91 are mostly an assertion that the LCA model is valuable rather than an explanation of why that is so. An LCA is a complex model and the choice of this model is not much discussed or justified relative to more straightforward approaches. You could have tested how each positive coping behavior related to each outcome (and, e.g., made an adjustment for multiple testing). Why didn’t you do that? You could then have issued more specific recommendations about which positive coping behaviors have which effects.

2. I think the manuscript should better address the question of reverse causality. You write about positive coping having effects on negative forms of coping, how do we know it is not the opposite? You write about positive coping reducing risk of mental health and substance use outcomes, how do we know it is not the opposite? Can you rule this out in any way and if not, how does it qualify your findings. You briefly acknowledge this possibility in lines 341-342 but the rest of the Discussion and the Abstract is written as if direction of causality has been established.

OTHER ISSUES

3. I think you should provide at least a brief characterization of sample recruitment in this manuscript. I do not think writing “Details about the study design and participant recruitment procedures can be found elsewhere” (line 112) is sufficient.

4. Please report the centering of variables in the product term for models with interaction (Table 4), since this affects interpretation.

5. Please report basic information about how you did the multiple imputation (e.g., number of imputations, specification of the chained equations).

6. Lines 270-273… should “drinking-to-cope motives” actually be on this list of significant differences between the latent classes? My reading of tables/figures is that they did not significantly differ on DMQ-coping.

7. FIGURE 2. Please add a note/legend. What are the error bars indicating, +/- 2 SEs? Can you indicate on here which contrasts are significant? The error bars around each mean do not answer that question.

8. Did you test any interactions with pandemic phase? Your observations cover a pretty large variety of pandemic contexts, from June 2020 to January 2022.

6. PLOS authors have the option to publish the peer review history of their article (what does this mean?). If published, this will include your full peer review and any attached files.

Reviewer #1: No

Reviewer #2: No

Reviewer #3: No

---

## [Author Response · Author response to Decision Letter 0]

17 Nov 2023

Please see uploaded document.

We have also added a non-author point of contact for data access requests, as requested.

---

## [Decision Letter · Decision Letter 1]

28 Dec 2023

Exercising healthy behaviors: A latent class analysis of positive coping during the COVID-19 pandemic and associations with alcohol-related and mental health outcomes

PONE-D-23-11488R1

Dear Dr. Ramchandani,

We’re pleased to inform you that your manuscript has been judged scientifically suitable for publication and will be formally accepted for publication once it meets all outstanding technical requirements.

Kind regards,

Giuseppe Marano

Academic Editor

PLOS ONE

Additional Editor Comments (optional):

Reviewers' comments:

Reviewer's Responses to Questions

**Comments to the Author**

1. If the authors have adequately addressed your comments raised in a previous round of review and you feel that this manuscript is now acceptable for publication, you may indicate that here to bypass the “Comments to the Author” section, enter your conflict of interest statement in the “Confidential to Editor” section, and submit your "Accept" recommendation.

Reviewer #1: All comments have been addressed

Reviewer #3: All comments have been addressed

2. Is the manuscript technically sound, and do the data support the conclusions?

Reviewer #1: (No Response)

Reviewer #3: Yes

3. Has the statistical analysis been performed appropriately and rigorously? 

Reviewer #1: (No Response)

Reviewer #3: I Don't Know

4. Have the authors made all data underlying the findings in their manuscript fully available?

Reviewer #1: (No Response)

Reviewer #3: No

5. Is the manuscript presented in an intelligible fashion and written in standard English?

Reviewer #1: (No Response)

Reviewer #3: Yes

6. Review Comments to the Author

Reviewer #1: (No Response)

Reviewer #3: **********************************************

The authors have responded adequately to my comments.

7. PLOS authors have the option to publish the peer review history of their article (what does this mean?). If published, this will include your full peer review and any attached files.

Reviewer #1: No

Reviewer #3: No

---

## [Editor Report · Acceptance letter]

5 Feb 2024

PONE-D-23-11488R1 

PLOS ONE

Dear Dr. Ramchandani, 

I'm pleased to inform you that your manuscript has been deemed suitable for publication in PLOS ONE. Congratulations! Your manuscript is now being handed over to our production team.

Kind regards, 

on behalf of

Dr. Giuseppe Marano 

Academic Editor

PLOS ONE